# Microbiome–Metabolomic Analysis Revealed the Immunoprotective Effects of the Extract of *Vanilla planifolia Andrew* (EVPA) on Immunosuppressed Mice

**DOI:** 10.3390/foods13050701

**Published:** 2024-02-26

**Authors:** Xin Zhang, Yunlong Li, Kexue Zhu, Chuan Li, Qingyun Zhao, Fenglin Gu, Fei Xu, Zhong Chu

**Affiliations:** 1Spice and Beverage Research Institute, Chinese Academy of Tropical Agricultural Sciences, Wanning 571533, China; 15595778045@163.com (X.Z.); li_yunlong_l@163.com (Y.L.); zhukexue163@163.com (K.Z.); qingyun_022@163.com (Q.Z.); xiaogu4117@163.com (F.G.); cz809@163.com (Z.C.); 2School of Food Science and Engineering, Hainan University, Haikou 570228, China; lichuan@hainanu.edu.cn; 3National Center of Important Tropical Crops Engineering and Technology Research, Wanning 571533, China; 4Key Laboratory of Processing Suitability and Quality Control of the Special Tropical Crops of Hainan Province, Wanning 571533, China; 5Sanya Research Institute of Chinese Academy of Tropical Agricultural Sciences, Sanya 572000, China

**Keywords:** extract of *Vanilla planifolia Andrew*, metabolomic analysis, immunomodulation, intestinal microbiota

## Abstract

This study investigated the immunoprotective effects of the extract of *Vanilla planifolia Andrew* (EVPA) on cyclophosphamide (Cy)-induced immunosuppression in mice. The results show that EVPA administration significantly alleviated the immune damage induced by Cy, as evidenced by an improved body weight, organ index, and colonic injury. A further analysis of microbial diversity revealed that the EVPA primarily increased the abundance of the beneficial bacteria *Verrucomicrobiota*, *Lactobacillaceae*, and *Lactobacillus* while decreasing *Akkermansiaceae*, *Akkermansia*, *Romboutsia*, and *Lactococcus*, thereby ameliorating the microbial dysbiosis caused by Cy. A metabolomic analysis revealed significant alterations in the microbial metabolite levels after EVPA treatment, including urobilinogen, formamidopyrimidine nucleoside triphosphate, Cer (d18:1/18:0), pantetheine, and LysoPC (15:0/0:0). These altered metabolites are associated with pathways related to sphingolipid metabolism, carbapenem biosynthesis, pantothenate and CoA biosynthesis, glycerophospholipid metabolism, and porphyrin metabolism. Furthermore, significant correlations were observed between certain microbial groups and the differential metabolites. These findings provide new insights into the immunomodulatory effects of EVPA on the intestinal microbiota and metabolism, laying the foundation for more extensive utilization.

## 1. Introduction

The intestines are crucial digestive and immune organs. The intestinal microbiota parasitize the host intestine and consist mainly of bacteria, archaea, fungi, and viruses. The diversity and complexity of the intestinal microbiota contribute to maintaining the stability of communications between the intestinal microorganisms and the host organism [1]. Studies have indicated that the intestinal microbiota facilitate the maturation of intestinal mucosal immunity, which affects the intestinal microorganisms and plays a role in the regulation of overall health [2].

Metabolites produced by the intestinal microbiota mediate communication between the host and microbiota [3]. Because of the substantial overlap in genomic content between the host and bacterial mitochondria, communication and regulation are induced between them [4]. This pathway is regulated by metabolites that facilitate the rapid response of immune cells to readjustment to maintain functionality [5]. Metabolites originating from the intestinal microorganisms play a crucial role in immune metabolism [6]. Deciphering all the internal and external changes resulting from the impact of microbiota on immune metabolism is a highly challenging endeavor.

High-throughput bacterial sequencing is a powerful tool for providing an overview of microbial communities and has advanced our understanding of the interactions between the host and intestinal microbiota [7]. Metabolomics provides information about the metabolism of microbes in the gut. Functional characteristic profiles of shared metabolites between the intestinal microbiota and the host have been obtained and found to be conducive to the understanding of the influence of diseases or drugs on the intestinal microbiota [8]. Combining metabolomics and metagenomics has facilitated a comprehensive interpretation of the interactions in gut metabolism [9]. This strategy has been widely used to elucidate the mechanisms of action of natural products on the host and intestinal microbiota.

Numerous modern diseases, such as inflammatory bowel and cardiovascular diseases, have been linked to immune dysfunction [10]. Therefore, the development of novel natural products as immune modulators has become a popular research topic. Natural products that have already been explored, such as polyphenols, polysaccharides, and peptides, exhibit the capacity to modulate intestinal immunity. Antimicrobial, antioxidant, anti-inflammatory, and immune-boosting properties are among the biological activities of the extracts of plants [11]. Extracts of plants regulate the immune system by stimulating the proliferation of immune-competent cells [12]. Chamaecyparis obtusa essential oil significantly inhibits the secretion of IL-25 and IL-33 by epithelial cells [13]. However, studies regarding the roles of the extracts of plants in intestinal immunity are scarce.

*Vanilla planifolia Andrew*, originally from Mexico, is a typical tropical economic crop. Fermented vanilla pods produce approximately 200 flavor compounds that give them their characteristic vanilla flavor [14]. Extract of *Vanilla planifolia Andrew* (EVPA) contains various aromatic compounds, with the primary constituent being vanillin. Additionally, 4-hydroxybenzaldehyde, vanillic acid, 4-hydroxybenzyl methyl ether, ethylvanillin, piperonal, and methyl anise are also present in EVPA [15]. EVPA exhibits in vitro antibacterial activity against *Enterobacter aerogenes* and *Streptococcus faecalis* [16]. Additionally, EVPA has a traditional medicinal use for the treatment of conditions such as colds, bronchitis, and asthma [17]. Vanillin, a simple phenolic compound, is the major component of EVPA. Guo et al. demonstrated that vanillin significantly alleviates obesity-related gut microbiota (GM) disorders including the decrease in alpha and beta diversity [18]. Monika Yadav et al. highlighted the human gut microbial features and metabolic bioprocess involved in vanillin catabolism to overcome its antimicrobial activity [19]. Vanillin was preliminary evaluated as quorum sensing inhibitors with well-documented additional beneficial effects on the host [20]. Studies on the changes in metabolites and their correlation with the microbiome after EVPA administration have not been reported. This study investigated the impact of EVPA on the immune regulation, colonic damage, microbial community, and metabolic characteristics of cyclophosphamide (Cy)-induced immunosuppressed mice to determine the immunoprotective effects of EVPA within the microbial metabolism axis.

## 2. Materials and Methods

### 2.1. Extraction of EVPA

The extraction of EVPA was conducted by referencing published methods with slight modifications and employing supercritical CO_2_ extraction technology [21]. The fermented pods (30% water content) were placed at a temperature of 50–55 °C, dried, crushed, and used. The extracted EVPA was then used for further experiments. The extraction conditions for HA640-70-27-C (2) supercritical CO_2_ extraction were as follows: extraction I heating temperature of 38 °C, CO_2_ flow rate of 83.92 L/h, separation I heating temperature of 58.1 °C, and extraction time of 80 min. The main components of EVPA are 4-methoxybenzyl alcohol (RC: 8.69%), coumarin (RC: 30.54%), and palmitic acid (RC: 4.83%). Our research group previously studied the main components of EVPA (Table 1) [21].

### 2.2. Animal Experiments

Male BALB/c mice (age: 8 weeks, 18–22 g) were acquired from Hunan Silaike Jingda Experimental Animal Co., Ltd. (Changsha, China). The certificate number is SCXK [Xiang] 2019–0004. The culture conditions included a room temperature of 23 ± 1 °C, relative humidity of 55 ± 10%, and light/dark cycle of 12 h/12 h. All animals underwent a one-week adaptation period before the experiment.

The total of 30 mice were randomly assigned to five groups: normal control group (NC, administered physiological saline), immunosuppression model group (MC, administered physiological saline), low-dose EVPA group (EVPAL, administered 50 mg/kg EVPA), medium-dose EVPA group (EVPAM, administered 100 mg/kg EVPA), and high-dose EVPA group (EVPAH, administered 200 mg/kg EVPA). The mouse models were intraperitoneally injected with Cy (100 mg/kg) for 3 days. The mice were subjected to oral gavage for 7 days. After the completion of the experiment, all experimental mice were anesthetized (5% chloral hydrate), and then the cervical vertebrae severed. Serum samples were immediately stored at −80 °C. After dissection, the spleens were weighed and preserved. Small intestines and colon tissues were preserved for subsequent experiments. This study adhered to national guidelines for the care and use of laboratory animals, and measures were taken to minimize the discomfort experienced by the animals during the experimental procedures. This study received approval from the Animal Ethics Committee of Hainan University (HNUAUCC-2021-00118).

### 2.3. Organ Index

The mouse spleens were excised and weighed accurately after euthanasia. The indices were calculated using the following formula:Organ-to-body weight ratio (mg/g) = organ weight (mg)/body weight (g)

### 2.4. Histopathological Observation

Following the washing step with physiological saline, the small intestinal tissue was fixed using a 10% formalin solution. Subsequently, the tissue underwent a series of procedures including dehydration, paraffin embedding, sectioning, and staining with hematoxylin and eosin (H&E). Samples were observed and photographed using an optical microscope.

### 2.5. High-Throughput Sequencing Analysis of 16S rDNA

Bacterial DNA from fecal samples was isolated utilizing the PureLink Microbiome DNA Purification Kit (A29790, Thermo Fisher Scientific, Waltham, MA, USA). The PCR amplification of the V3–V4 region of the 16S rRNA gene was carried out for each sample using the specific primers 338F and 806R. The resulting PCR products were subjected to sequencing on the Illumina MiSeq-PE250 platform to obtain the microbial diversity information.

### 2.6. UPLC–Q-TOF-MS Analysis of Mice Feces

Fecal samples (100 mg) were removed from the refrigerator and weighed, and then added to a mixture consisting of methanol/acetonitrile/water in a 4/4/2 ratio. The chromatographic column used was a Waters ACQUITY UPLC HSS T3 C18 (2.1 mm × 100 mm, 1.8 µm). The gradient elution program and mass spectrometry acquisition conditions were as previously described [22].

The collected data were imported into SIMCA-P for analysis, employing principal component analysis (PCA) and orthogonal projections to latent structures discriminant analysis (OPLS-DA). To assess the quality of the OPLS-DA model, R2X, R2Y, and Q2 were utilized as evaluation metrics. Furthermore, in order to identify potential small-molecule compounds, the fold change (FC) threshold (FC ≥ 2 and FC ≤ 0.5), along with the variable importance in projection (VIP) value (VIP > 1), were employed as screening indicators within the validated OPLS-DA model. Characteristic small molecules were identified through large-scale searches and manual verification. A pathway analysis of potential biomarkers was conducted using MetOrigin [23]. The Kyoto Encyclopedia of Genes and Genomes (KEGG), the Human Metabolome Database (HMDB), Biochemical Genetic and Genomic (BiGG) knowledgebase, Chemical Entities of Biological Interest (ChEBI), Food Database (FoodDB), DrugBank, and Toxin and Toxin Target Database (T3DB) were used to analyze metabolic pathways and elucidate the molecular mechanisms underlying substance changes.

### 2.7. Statistical Analysis

The means ± standard error of the mean (SEM) were employed to present the experimental results. In utilizing SPSS 26.0 software, statistical analyses were conducted, and Origin 2023 was utilized to generate graphical representations. One-way analysis of variance (ANOVA), followed by the Duncan’s multiple range test, was performed to determine the significance of differences among various groups. Statistically significant differences were indicated by *p* < 0.05. In the figures and tables, distinct superscript letters are used to indicate significant differences. Additionally, for the analysis of standardized mass spectrometry data, the SIMCA software (version 14.1, Umetrics AB, Umea, Sweden) was utilized.

## 3. Results

### 3.1. EVPA Improved the Body Weight of Immunosuppressed Mice

The mice were fed a standard diet before Cy injection. Three days after Cy injection, the vitality and food intake of the mice decreased. The fur of the mice became dull and lacked luster, and their mental states weakened. However, after gastric perfusion with different doses of EVPA, the symptoms in mice were alleviated. Figure 1A demonstrates the influences of different doses of EVPA on the weight of immunosuppressed mice induced by Cy. The results demonstrate that a mouse’s body weight in the NC group exhibited an upward trend during the experiment. In contrast, a mouse’s body weight in the MC group significantly decreased after Cy injection, confirming the successful establishment of the model. After seven consecutive days of gastric perfusion with EVPA, the EVPAM and EVPAH groups exhibited significant weight recovery, indicating that medium to high doses of EVPA alleviated the damage to immunosuppressed mice induced by Cy.

### 3.2. EVPA Improved the Spleen Index in Immunosuppressed Mice

The spleen is the primary immune organ of the human body. The spleen index is an indicator of immune organ function [24]. The spleen index of the MC group was observed to be significantly higher than that of the NC group (Figure 1B). This observation suggests that Cy caused splenomegaly in mice. These results are consistent with previous findings [25]. The spleen index of immunosuppressed mice significantly decreased with different doses of EVPA. This result indicates that EVPA restored the damage to the structure and function of the spleen.

### 3.3. Effects of EVPA on Histological Changes in the Small Intestine

H&E staining is commonly used to observe pathological tissues and examine their morphological changes [26]. As shown in Figure 1D, the intestinal villus structure of the NC-group mice was intact, and the mucosal surface structure was complete and smooth. Nonetheless, the colonic mucosa in the MC group suffered severe damage, with partial villus swelling, shortening, and ulcers. After intervention with EVPA, the morphology of the mouse colonic tissues improved, and the integrity of the mucosa was effectively restored. In the EVPAM and EVPAH groups, the villus structure in the small intestine remained intact, showing considerable improvement.

### 3.4. Effect of EVPA on the Intestinal Flora of the Immunosuppressed Mice

The intestinal microflora primarily affect the occurrence and development of immune-related diseases by modulating mucosal immunity [1]. Alpha diversity reflects the abundance and species diversity of individual samples. Compared to the NC group, Cy-induced immunosuppressed mice exhibited a significant decrease in Shannon’s diversity index and a significant increase in Simpson’s diversity index (Appendix A). After intervention with EVPA, the Ace and Chao1 indices were significantly decreased in immunosuppressed mice. These results suggest that Cy caused a decrease in the intestinal microbiota diversity. Furthermore, after intervention with EVPA, the intestinal microbiota richness was reduced.

β diversity is primarily used to describe changes in species composition at the spatial scale and evaluate between-group similarity. As shown in Figure 2A,B, samples from the different treatment groups clustered separately, and samples in the NC and MC groups were distinctly separated. This result indicates an evident alteration in the MC group mouse’ intestinal microbiota. Samples from the groups treated with different concentrations of EVPA were distinctly separated from the MC group. Further, the samples from the EVPAH groups were closer to those from the NC group.

The proportional abundance of each species was evaluated based on the community distribution at different taxonomic levels (phylum, families, and genera). As shown in Figure 2C, the microbial structure of the fecal samples was mainly composed of *Firmicutes*, *Verrucomicrobiota*, *Bacteroidota*, *Cyanobacteria*, *Actinobacteriota*, and *Campilobacterota* at the phylum level. Among them, *Firmicutes*, *Verrucomicrobiota*, and *Bacteroidota* were the dominant flora, with their relative abundances accounting for over 90%. Compared to the NC group, in the MC group, the abundance of Firmicutes noticeably decreased and the abundance of *Verrucomicrobiota* increased. Similar results were reported by Liu et al. [27]. After EVPA treatment, the gut microbial structure showed similarity to that of the NC group, particularly in the EVPAM and EVPAH groups. This result indicates that Cy had an obvious effect on the composition of the intestinal microbiota at the phylum level, and EVPA effectively improved the structure of the intestinal microbiota.

In addition, alterations in the composition of the intestinal microbiota at the family level were examined (Figure 2D). *Lactobacillaceae*, *Akkermansiaceae*, *Muribaculaceae*, *Lachnospiraceae*, and *Bacteroidaceae* were the predominant intestinal microbiota at the family level. The relative abundance of *Akkermansiaceae* increased significantly in the MC group, whereas that of *Lactobacillaceae* decreased significantly compared to the NC group. Compared to the MC group, in the EVPAM and EVPAH groups, the relative abundance of beneficial microorganisms, such as *Lactobacillaceae*, increased, while that of *Akkermansiaceae* decreased. These results indicate that Cy affects the intestinal microbiota composition in mice. Following EVPA treatment, the relative abundance of beneficial bacterial families increased, leading to the regulation of dysbiosis in the intestinal microbiota.

The composition of the intestinal microbiota was analyzed and displayed at the genus level by clustering between groups. The results of a cluster analysis revealed that the dominant bacteria in the NC group were similar to those in the EVPAM and EVPAH groups and differed from those in the MC group (Figure 2E). The relative abundance of *Lactobacillus* in the MC group was reduced, whereas the relative abundances of *Akkermansia*, *Romboutsia*, *Helicobacter*, and *Lactococcus* increased compared to the NC group. After intervention with EVPAH, the relative abundance of the bacterial genera was reversed. The linear discriminant analysis effect size (LEfSe) was used to estimate the impact of each component’s abundance on the observed differential effect. As illustrated in Figure 2F, LEfSe analysis demonstrated that *Gordonibacter*, *Staphylococcus*, *Rikenellaccac-RC9-gut-group*, *Butyricimonas*, *Bacteroides*, *Anacrofustis*, and *Parabacteroides* were the main intestinal microbiota in the NC group. The main intestinal microbiota of the MC group were *Akkermansia*, *Erysipelatoclostridium*, and *Blautia*. The main microbiota in the EVPA group were *Prevotellaceae_NK3B31* and *Lactobacillus*. *Lactobacillus* was the predominant genus in the EVPAH group (LDA = 5.46). These results indicate that Cy reduces the diversity of the intestinal microbiota in mice and that EVPAH primarily regulates intestinal immunity by increasing the abundance of *Lactobacillus*.

### 3.5. Effects of EVPA on Metabolite Composition in Immunosuppressed Mice

Metabolites are considered bridges between genotypes and phenotypes. The investigation of metabolite levels and gene functions enhance our understanding of biochemical and molecular mechanisms.

As shown in Figure 3A,B, the principal component analysis results of mouse fecal samples indicated a clear separation between the MC and NC groups, particularly in the positive ion mode. The results of the EVPA-treated groups tended to approach those of the NC group. Similarly, the PLS-DA results indicate clear clustering of the samples from each group (Figure 3C,D). The results of the EVPA-treated groups (especially the EVPAH group) are similar to those of the NC group. These findings further validate the alterations in endogenous metabolites induced by Cy and demonstrate the effective regulation of Cy-induced metabolic disorders in mice through EVPA administration. To elucidate the effect of EVPA on immunosuppressed mice, fecal sample data were analyzed using the OPLS-DA model. The score plots of the OPLS-DA model are illustrated in Figure 4A–H, indicating that the MC group was effectively distinguished from the NC and EVPA groups.

The OPLS-DA scoring parameters for the groups are presented in Table 2. A Q^2^ value > 0.5 suggests that EVPA mitigates the disruption of mouse fecal metabolites induced by Cy. The samples from each group were completely separated, indicating distinct characteristics of the mouse fecal metabolic profiles. Similarly, the results of 200 permutation tests revealed that none of the OPLS-DA models exhibited overfitting (Appendix A).

Potential variables that contributed to group separation were selected using the VIPs and FCs. Differential metabolite identification was performed through a comparison with online databases. Ultimately, 18 differential metabolites were selected from the fecal samples (Figure 5). After Cy intervention, the levels of 7-sulfocholic acid, 3-pyridylacetic acid, cyclopassifloic acid B, ganglioside GD1a (d18:1/22:0), ceramide (Cer) (d18:1/18:0), N-arachidonoyl tyrosine, N-oleoyl tyrosine, phosphatidic acid (PA) (22:6/20:5), PA (20:0/22:4), lysophosphatidylcholine (LysoPC) (15:0/0:0), heme, and pantetheine in the feces increased, whereas urobilinogen levels decreased. However, the levels of these compounds were reversed through EVPA treatment. These results suggest that the injection of Cy led to metabolic disruptions in the mouse gut, and EVPA intervention modulates these metabolic disturbances.

### 3.6. Effect of EVPA on the Metabolic Pathways of the Intestinal Microbiota

Metabolites from the intestinal microbiota are essential intermediaries that disrupt the host microbiota metabolism. Differential metabolites were classified and selected based on various databases (HMDB, KEGG, BiGG, ChEBI, FoodDB, and DrugBank). These metabolites were categorized into six groups according to their source: host (mammalian), microbiota (bacterial), food (nutrition and plants), drugs, and environment (toxins and pollutants). Among these metabolites, five were host-related, seven were microbiota metabolism-related (with five being co-metabolites of the host and microbiota), twelve were related to food, two were associated with drugs, and five were metabolites of unknown functions (Figure 6A). The host-microbiota co-metabolites included urobilinogen, formamidopyrimidine nucleoside triphosphate, Cer (d18:1/18:0), pantetheine, LysoPC (15:0/0:0), siroheme, and trichostatin A. Among these, the host-microbiota co-metabolites comprised urobilinogen, formamidopyrimidine nucleoside triphosphate, Cer (d18:1/18:0), and pantetheine.

Metabolic Pathway Enrichment Analysis (MPEA) is a method that combines enrichment analysis with pathway topological features to determine the most relevant pathways. The MPEA has been widely used in metabolomic studies. MPEA analysis identified five metabolic pathways that were significantly affected by EVPA, including four host-microbiota co-metabolic pathways (sphingolipid metabolism, carbapenem biosynthesis, pantothenate and CoA biosynthesis, and glycerophospholipid metabolism) and one microbial metabolic pathway (porphyrin metabolism) (Figure 6B). These results suggest that Cy disrupts the distribution of the microbial metabolic spectrum and high-dose EVPA treatment improves metabolic disorders in immunosuppressed mice.

### 3.7. The correlation between Intestinal Microbiota and Metabolites

The relationship between the microbiota and metabolites was explored using Pearson’s correlation analysis. The results shown in Figure 6C,D indicate that the levels of host–microbiota co-metabolites, including LysoPC (15:0/0:0), GD1a (d18:1/22:0), Cer (d18:1/18:0), N-arachidonoyl tyrosine, pantetheine, siroheme, and 7-sulfocholic acid, showed a positive correlation with *Lactobacillaceae*, and negative correlations with *Akkermansiaceae*, *Muribaculaceae*, and *Prevotellaceae* at the family level. At the genus level, the levels of these metabolites positively correlated with the relative abundance of *Lactobacillus* and negatively correlated with *Akkermansia*, *Parabacteroides*, *Ruminococcus*, and *Mucispirillum*. These results indicate that changes in the intestinal microbiota structure are linked to intestinal metabolic disruption, and that EVPAH treatment modulates the intestinal microbiota in immunosuppressed mice, thereby regulating intestinal metabolism.

## 4. Discussion

Cy is used extensively to treat various tumors. However, the continuous administration of Cy leads to side effects, including immunosuppression, intestinal barrier impairment, oxidative stress, and the dysregulation of the intestinal microbiota [28]. In this study, we established a Cy-induced immunosuppressed mouse model to investigate the effects of EVPA in immunosuppressed mice. The immunosuppressed mice exhibited significant symptoms, including a noticeable reduction in body weight and an increase in the spleen index. Additionally, histopathological observations revealed damage to the colonic mucosa of mice. These results suggest that Cy intervention impaired the immune response and intestinal barrier in mice, which is consistent with previous research [29]. After EVPA intervention, especially in the EVPAH group, there was a significant improvement in the immune dysregulation symptoms and colonic damage induced by Cy in the immunosuppressed mice. EVPA has been demonstrated to show excellent antibacterial activity in vitro [30]. Shin found that the essential oil extracted from *Chamaecyparis obtusa* possessed antibacterial, antiallergic, and anti-inflammatory properties [31].

The gut microbiome is considered a significant environmental factor and an effective strategy for alleviating immune diseases and preventing immune-related disorders [32]. Cy treatment altered the diversity and composition of the intestinal microbiota, increased the relative abundance of *Akkermansiaceae*, and induced microbial dysbiosis. *Akkermansiaceae* belongs to a family of mucin-degrading bacterial species. SARS-CoV-2-infected mice exhibited severe microbiome damage, increased intestinal permeability, and elevated levels of *Akkermansiaceae* [33]. Intervention with EVPA primarily manifested in an increased relative abundance of *Lactobacillaceae*. *Lactobacillaceae* are common probiotics that enhance intestinal immunity and maintain gut health. Litsea cubeba essential oil regulates inflammation by modulating the relative abundance of *Lactobacillaceae* [34].

The host and microbiota have co-evolved over the long term, and the intestinal microbiota actively contribute to the regulation of host physiology and immunity. The glycerophospholipid and sphingolipid metabolisms were involved in the relevant pathways of EVPA-mediated immune regulation. Cer (d18:1/18:0), a ceramide, is a major metabolite of the sphingolipid metabolic pathway. The sphingolipid metabolic pathway is an important target for the treatment of many diseases [35]. Ceramide is considered a key factor in mediating and regulating the functions of various immune cells, such as modulating immune cell responses to bacteria, viruses, and other foreign pathogens as well as the production of cytokines [36]. GD1a (d18:1/22:0), a ganglioside, is a component of the cellular membrane that plays a crucial role in the regulation of cell signal transduction [37]. The levels of Cer (d18:1/18:0) and GD1a (d18:1/22:0) significantly increased in mice with immunosuppression induced by Cy, and EVPA treatment restored this disruption. The glycerophospholipid metabolism may be a key pathway involved in the immune regulatory mechanism of EVPA. In this study, significant changes in glycerophospholipids (including PA, PC, and PE) were observed in the feces of immunosuppressed mice. Glycerophospholipids are involved in various immune-mediated diseases, and the high expression of glycerophospholipids in the feces of immunosuppressed mice not only suggests cell membrane damage, but also indicates immune activation [38]. Previous studies have shown that LysoPCs suppress the secretion of proinflammatory cytokines and are inversely correlated with IL-1β, IL-6, and TNF-α [39]. Cao et al. revealed a significant correlation between the relative abundance of *Lactobacillus* and ceramide levels in cyclophosphamide-induced immunosuppressed mice [36].

Pantetheine is an analog of pantothenic acid (Vitamin B5), which serves as an intermediate in CoA synthesis [40]. CoA is crucial for the transfer of acetyl groups into the TCA cycle. In this study, we found that EVPA regulates pantetheine levels. Additionally, there was a positive correlation between pantetheine content and the relative abundance of *Lactobacillus*. Previous studies have indicated that the *Lactobacillus bulgaricus* factor (present in both pantetheine and pantetheine forms) plays a crucial role in stimulating the growth of *L. bulgaricus* [41]. These findings suggest that EVPA may alleviate gut dysbiosis and immune dysregulation by modulating *Lactobacillus* and its metabolite (pantetheine), which affects the carbapenem, pantothenate, and CoA biosynthesis pathways.

The porphyrin biosynthetic pathway is present in almost all eukaryotic cells. Hemoglobin is a crucial component of electron transport chain proteins and is involved in a variety of oxidative metabolic processes within biological cells, including oxygen transport [42]. Siroheme is a heme cofactor of sulfite and nitrite reductases that participate in the formation of ammonia and sulfides in microorganisms [43]. Bilirubin (produced by heme metabolism) is degraded by intestinal bacteria to produce urobilinogen, with the majority being excreted in the feces. In this study, Cy induced metabolic disruption in mice, leading to an increase in heme content and a decrease in urobilinogen content in the feces. Intervention with EVPAH normalized the levels of these metabolites. *Clostridium* and *Bacteroides* have been demonstrated to metabolize bilirubin to urobilinogens. However, these specific bacterial species were not identified in our study. These findings illustrate the intricate interactions between the intestinal microbiota and host. EVPAH exhibits a positive regulatory role in rectifying the gut microbial imbalance and intestinal metabolic disruption induced by Cy.

In summary, EVPAH demonstrated immunoprotective effects in immunosuppressed mice, which were related to the modulation of intestinal microbiota and the improvement of microbial metabolites to alter intestinal immunity. These findings provide new insights into the protective role of EVPA in the immune system.

## 5. Conclusions

This study explored the immunoprotective effects of EVPA in Cy-induced immunosuppressed mice and revealed that EVPAH significantly improved body weight, organ indices, and colon damage. The microbiota diversity analysis indicated that EVPAH primarily exerted immune-modulating effects by increasing the relative abundance of *Lactobacillus* and suppressing harmful bacteria. The metabolomic analysis demonstrated that EVPAH mitigated intestinal metabolic disruption and enhanced gut immunity by regulating gut microbial metabolites. Furthermore, Pearson’s correlation analysis revealed a strong association between changes in the intestinal microbiota structure and variations in microbial metabolite content. These findings suggest that EVPAH intervention collectively reshaped the immune function of the immunosuppressed mice by reconstructing the intestinal microbiota and microbial metabolites. These results provide insights into the immunomodulatory effects of EVPA from a microbiome–metabolome perspective. Subsequent studies will be conducted to isolate the specific components of EVPA and investigate specific microbial communities and metabolites.

## Figures and Tables

**Figure 1 foods-13-00701-f001:**
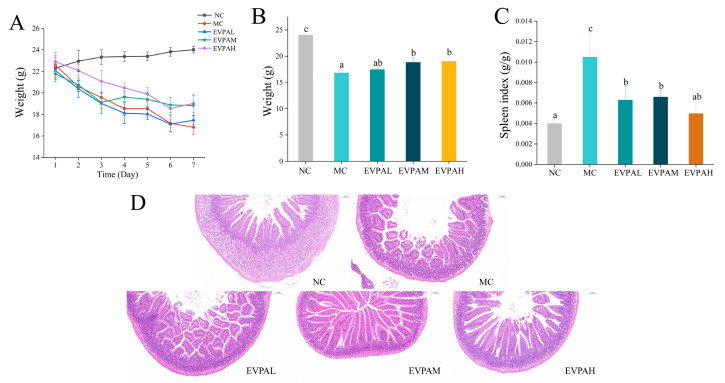
Effects of EVPA on weight, immune organ indexes, and the histology of the colons of immunosuppressed mice. (**A**) Changes in body weight; (**B**) Body weight on Day 7; (**C**) Spleen index; (**D**) H&E-stained colon tissues (100× magnification). Values with different letters of the same indicator are significantly different.

**Figure 2 foods-13-00701-f002:**
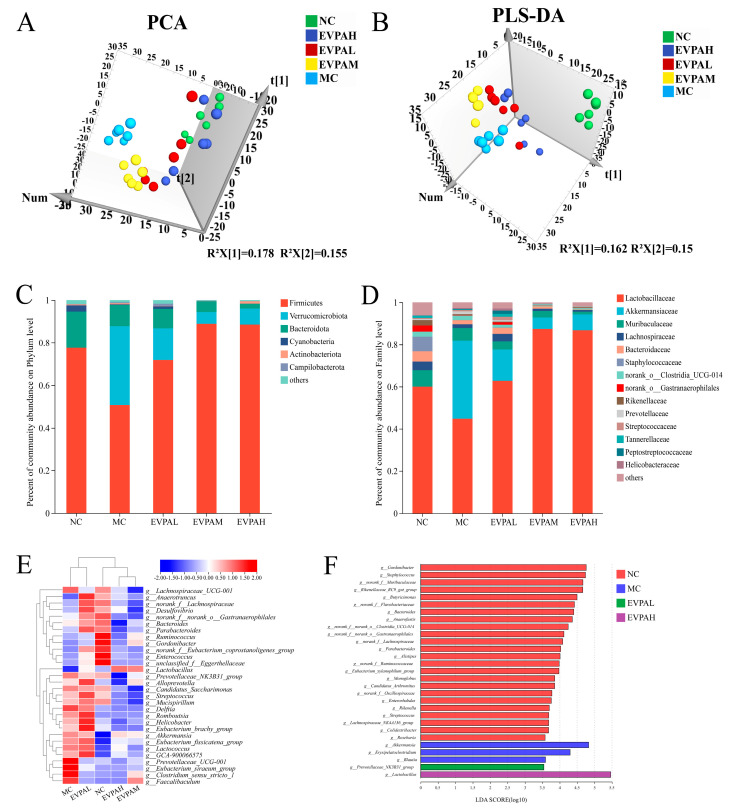
Effects of EVPA on the gut microbiota in immunosuppressed mice. (**A**) PCA of gut microbiota; (**B**) PLS-DA of gut microbiota; (**C**) Microbial community stack column plot at the phylum level; (**D**) Microbial community stack column plot at the family level; (**E**) Heatmap of the top 30 dominant genera in each group; (**F**) Comparisons of the microbial variance through a LEfSe analysis on the genus. Values with different letters are significantly different.

**Figure 3 foods-13-00701-f003:**
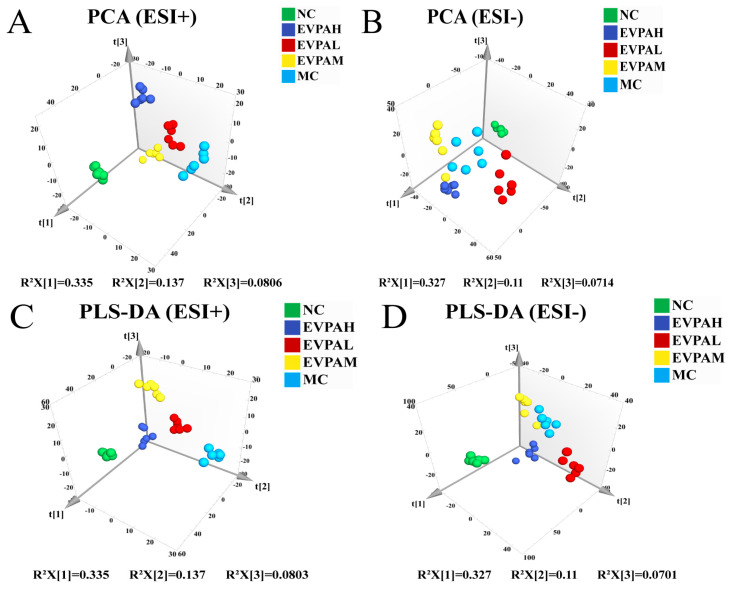
Effects of EVPA on the fecal metabolic profile in Cy-induced immunosuppressive mice. PCA plots of metabolites among the NC, MC, and EVPA groups in the (**A**) positive and (**B**) negative ion modes; PLS-DA plots of metabolites among the NC, MC, and EVPA groups in the (**C**) positive and (**D**) negative ion modes. PCA, PLS-DA, and OPLS-DA are metabolomic analysis methods commonly used to identify metabolic profile differences between groups.

**Figure 4 foods-13-00701-f004:**
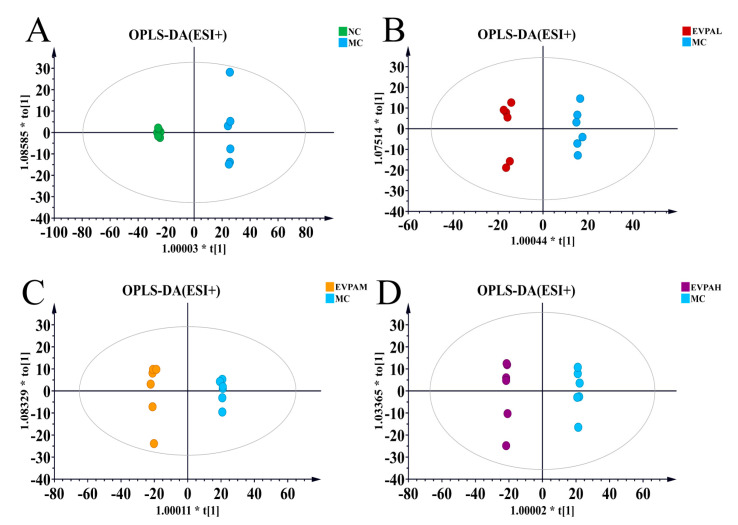
OPLS-DA scores of mouse fecal samples: (**A**–**D**) positive ion modes; (**E**–**H**) the negative ion modes. OPLS-DA is a metabolomics analysis method commonly used to screen differential metabolites between groups.

**Figure 5 foods-13-00701-f005:**
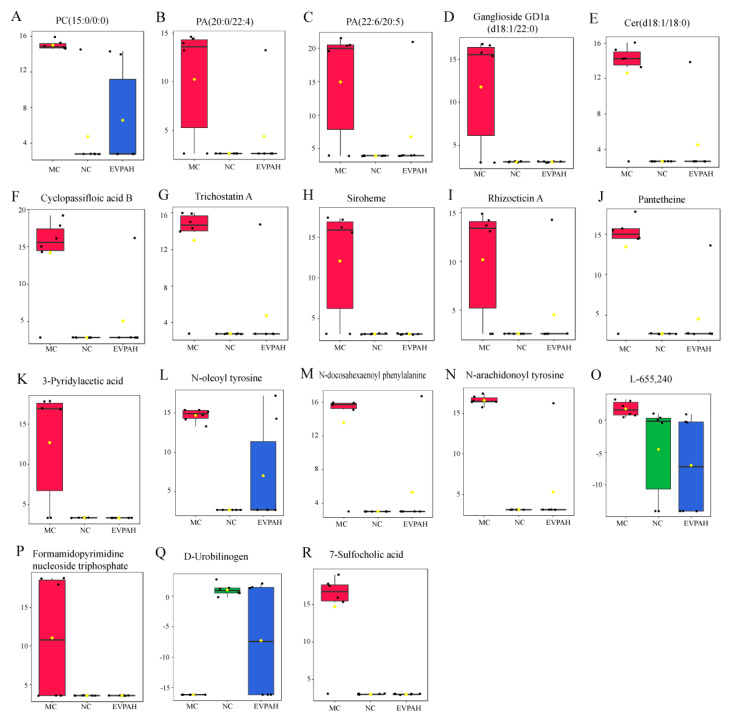
Box plots of the relative concentrations of differential metabolites (biomarkers) in the NC, MC, and EVPAH groups. The relative concentrations of (**A**) phosphatidylcholine (PC) (15:0/0:0), (**B**) phosphatidic acid (PA) (20:0/22:4), (**C**) PA (22:6/20:5), (**D**) ganglioside GD1a (d18:1/22:0), (**E**) ceramide (Cer) (d18:1/18:0), (**F**) gyclopassifloic acid B, (**G**) trichostatin A, (**H**) siroheme, (**I**) rhizocticin A, (**J**) pantetheine, (**K**) 3-pyridylacetic acid, (**L**) N-oleoyl tyrosine, (**M**) N-docosahexaenoyl phenylalanine, (**N**) N-arachidonoyl tyrosine, (**O**) L-655,240, (**P**) formamidopyrimidine nucleoside triphosphate, (**Q**) urobilinogen, and (**R**) 7-sulfocholic acid.

**Figure 6 foods-13-00701-f006:**
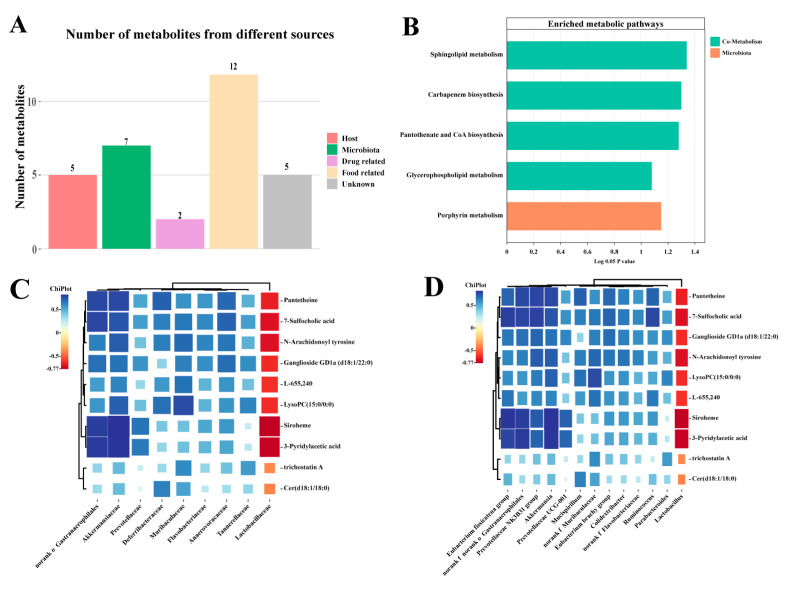
Metabolic pathways and correlation analysis of EVPA in immunosuppressed mice. (**A**) Functional screening of fecal metabolites; (**B**) Pathway analysis of metabolic pathways of differential metabolites; (**C**) Correlation between microbial and fecal metabolites at the phylum level (**C**) and genus level (**D**).

**Table 1 foods-13-00701-t001:** The main composition of the extract of *Vanilla planifolia Andrew* (EVPA).

Number	Compound	Molecular Formula	Relative Content (%)
1	Styrene	C_8_H_8_	0.23 ± 0.14
2	2,2,4,6,6-pentamethyl-heptane	C_12_H_26_	0.15 ± 0.11
3	4-methoxy-benzaldehyde	C_8_H_8_O_2_	0.14 ± 0.10
4	4-methoxy-benzenemethanol	C_8_H_10_O_2_	8.69 ± 2.13
5	4-methoxy-benzoicacidmethylester	C_9_H_10_O_3_	0.17 ± 0.12
6	3-phenyl-2-Propenoicacidmethylester(E)-	C_10_H_10_O_2_	0.03 ± 0.01
7	Vanillin	C_8_H_8_O_3_	30.54 ± 1.59
8	Caryophyllene	C_15_H_24_	0.10 ± 0.08
9	4-methoxy-benzenemethanolacetate	C_10_H_12_O_3_	0.18 ± 0.11
10	4-methoxybenzoicacid	C_8_H_8_O_3_	0.84 ± 0.08
11	Dodecanoic acid	C_12_H_24_O_2_	0.06 ± 0.01
12	Myristic acid	C_14_H_28_O_2_	0.08 ± 0.05
13	2-Pentadecanone,6,10,14-trimethyl-	C_18_H_36_O	0.12 ± 0.06d
14	Hexadecanoic acid, methyl ester	C_17_H_34_O_2_	0.12 ± 0.07d
15	n-Hexadecanoic acid	C_16_H_32_O_2_	4.83 ± 0.12
16	Hexadecanoicacid, ethylester	C_18_H_36_O_2_	0.15 ± 0.07
17	9,12-Octadecadienoicacid(Z,Z)-,methyl ester	C_19_H_34_O_2_	0.65 ± 0.26
18	11-Octadecenoicacid, methyl ester	C_19_H_36_O_2_	0.17 ± 0.11
19	Linoleic acid	C_18_H_32_O_2_	3.81 ± 2.59
20	9,12-Octadecadienoicacid, ethyl ester	C_20_H_36_O_2_	1.16 ± 0.58
21	Oleic Acid	C_18_H_34_O_2_	0.73 ± 0.45
22	Heneicosane	C_21_H_44_	0.56 ± 0.06
23	Tetracosane	C_24_H_50_	1.39 ± 0.98
24	Nonadecane-2,4-dione	C_19_H_36_O_2_	1.06 ± 0.77
25	1-Octacosanol	C_28_H_58_O	1.58 ± 0.56
26	Octacosane	C_28_H_58_	4.31 ± 0.35
27	Heptacosane	C_27_H_56_	3.43 ± 0.44

**Table 2 foods-13-00701-t002:** OPLS-DA scoring parameters between groups.

Model	Group Comparison	R^2^ Y (cum)	Q^2^ (cum)
ESI+	MC VS NC	0.999	0.972
ESI+	MC VS EVPAL	0.996	0.880
ESI+	MC VS EVPAM	0.998	0.960
ESI+	MC VS EVPAH	1	0.962
ESI−	MC VS NC	0.999	0.956
ESI−	MC VS EVPAL	0.993	0.808
ESI−	MC VS EVPAM	0.998	0.776
ESI−	MC VS EVPAH	0.997	0.851

## Data Availability

The original contributions presented in the study are included in the article/Appendix A; further inquiries can be directed to the corresponding author.

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
