# Peer review of "Microbiome–Metabolomic Analysis Revealed the Immunoprotective Effects of the Extract of Vanilla planifolia Andrew (EVPA) on Immunosuppressed Mice"

_foods, 2024, doi:10.3390/foods13050701_

Round 1

Reviewer 1 Report

Comments and Suggestions for Authors

The study of orchids is a very promising topic. Therefore, we can only welcome the publication of another article on the topic of Vanilla orchid extract. The authors conducted a fairly large and high-quality study. The results will be of interest not only to practicing gastroenterologists, but also to specialists in the field of microbiology, biochemistry, and physiology. The authors presented the research results quite well, but some issues need to be corrected.

Key notes:

1. In the name, it is better to replace "Vanilla orchid extract (VOE)" with an "extract of the Latin name of the plant".

2. The abstract is not informative enough. In a Scopus or Web of Science database, it will not attract enough readers. You need to add the same amount of text to the abstract, making sure to mention as many names of microorganisms and chemical substances as possible.

3. It is better not to write “(p < 0.05)” twice in the annotation. All judgments in the annotation must be statistically reliable even without this.

4. I recommend adding to the Materials and Methods a table with the names, structural formulas and % content of 20-25 substances in the extract. I understand that the authors did not do this research themselves, however, the literature sources can be mentioned in the table, this will be enough.

5. All fonts in all figures should be no larger (for example, Figure 1, 3) and no smaller (for example, Figure 5) than the size of the fonts in the main text of the article. There is no need to use bold font in pictures.

6. Each drawing must be self-sufficient. For example, reading the title of Figure 3 does not allow the reader to understand what exactly the authors depicted in it without reading the Materials and Methods section. Authors should try harder to ensure that readers understand as many facts and ideas as possible that they want to convey to readers through drawings.

7. In Figure 1e, in the photographs, you need to mark the most noticeable elements (types of cells) with numbers 1-7, and decipher their names in the title of the picture. In the bottom corner of each photo you need to draw a 10 or 50 micron bar and describe it in the title of the picture.

8. In Figure 1a, “± standard error” are superimposed; the reader will not understand which curve the standard error refers to.

9. In Figures 2e and 2f, the names of genera and species (but not the names of families and phyla) should be written in italics.

10. I think Figure 2e is the most significant result of this paper. I recommend adding it to the graphical abstract of this article. Only in the graphic annotation it is necessary to write the name of the experiment option in full. This will greatly increase the citation potential of this article.

11. It is likely that each box in Figure 4 is reliably different from each other. But readers cannot understand this from the image or from the title of the drawing. This needs to be corrected. Probably all the substances in the title of this figure need to be named correctly so that readers understand what we are talking about without reading other sections of this article. The names of substances do not need to be capitalized everywhere.

12. Probably, Figure 5d should also be placed in a graphic abstract, describing all its elements quite fully.

13. For articles in the reference list, it is better to indicate the DOI indices.

14. I do not recommend significantly expanding or changing the text part of the article (everything is written quite clearly and clearly). I highly recommend taking a closer look at each drawing. It is not a problem for a magazine to enlarge each image by 50-100%, even if this increases the length of the article by 8 pages. The main thing is that each image is clear to readers, so that the authors’ ideas are perceived by readers. One drawing cannot occupy more than one page. It would probably be advisable to split some of the drawings into two and give them their own names.

Reviewer 2 Report

Comments and Suggestions for Authors

The results presented are interesting although the writing and/or presentation could improve.

Line 170. In the text Fig. 1C is mentioned but it should say Fig. 1D.

Figure 1A. It is recommended to normalize each of the kinetics with respect to the baseline so that the change can be appreciated from the same point (all kinetics starting at 1). On quick review of the figure, VOEH appears to have no effect.

It stands out that from day 5-6 everyone loses weight, as if they had run out of food. The weight recovers from 6 to 7 because they eat more?

Fig. 1B is the final weight on day 7? It is recommended to graph the weight change from day 1 to day 7 in each treatment.

Fig. 1C, in addition to the index, it is recommended to include the histology of the spleen to appreciate the differences as occurs with the colon (Fig. 1D).

Line 223. The wording is typical of the Discussions and not of the Results section.

Figure 2D. It highlights the recovery of some genres but not the original diversity of control. A possible reason is not explained.

Line 272. "fcecal"?

Line 320. Paragraph cut...check paragraph format.

Comments on the Quality of English Language

Language and style review recommended

Reviewer 3 Report

Comments and Suggestions for Authors

1- line 71, need more characterization of pharmacological effect.

2- line 74, vanillin, should be described more and its activity mentioned.

3-  line .88, difference between fermented and non-fermented plant should be mentioned. 

4- cyclophosphamide (Cy)-induced immunosuppression model is not well defined

6-  I received two word files, both have the same content.  they should be revisedand resend by the authors.   

Comments on the Quality of English Language

were written above. 

general revision
